# The Absence of Bovine Serum Albumin (BSA) in Preimplantation Culture Media Impairs Embryonic Development and Induces Metabolic Alterations in Mouse Offspring

**DOI:** 10.3390/ijms26146989

**Published:** 2025-07-21

**Authors:** Jannatul Ferdous Jharna, Md Wasim Bari, Norermi Firzana Alfian, Satoshi Kishigami

**Affiliations:** 1Department of Integrated Applied Life Science, Integrated Graduate School of Medicine, Engineering, and Agricultural Sciences, University of Yamanashi, Yamanashi 400-8510, Japan; g24diba2@yamanashi.ac.jp (J.F.J.); g24dib02@yamanashi.ac.jp (N.F.A.); 2Faculty of Life and Environmental Sciences, University of Yamanashi, Yamanashi 400-8510, Japan; 3Center for Advanced Assisted Reproductive Technologies, University of Yamanashi, Yamanashi 400-8510, Japan; 4Advanced Biotechnology Center, University of Yamanashi, Yamanashi 400-8510, Japan

**Keywords:** embryo culture, bovine serum albumin, preimplantation, metabolic programming, DOHaD, glucose intolerance

## Abstract

Bovine serum albumin (BSA), the most commonly used protein in preimplantation embryo culture media, performs a variety of physiological functions. However, its involvement in long-term effects remains largely unclear. To investigate its physiological importance in culture media, we examined the developmental and metabolic consequences of BSA deprivation during preimplantation stages in mice. Embryos cultured in BSA-free media during specific time windows exhibited impaired blastocyst formation, with continuous deprivation from the two-pronuclei (2PN) stage significantly reducing trophectoderm (TE) and inner cell mass (ICM) cell numbers (*p* < 0.05), indicating compromised viability. Short-term BSA deprivation similarly disrupted lineage allocation, underscoring the sensitivity of early embryos to nutrient availability during cell fate determination. Although birth rates remained unaffected, suggesting compensatory mechanisms, longitudinal analysis revealed sex-specific metabolic dysfunction. Male offspring developed progressive glucose intolerance by 16 weeks, exhibiting elevated fasting glucose levels (*p* < 0.05) and impaired glucose clearance, whereas females showed no significant alterations in glucose metabolism. This study demonstrates that protein restriction during the preimplantation period not only disrupts early embryonic development but also programs long-term metabolic dysfunction, underscoring the importance of optimizing culture conditions in assisted reproductive technologies to minimize future health risks.

## 1. Introduction

Advancements in embryo culture systems have greatly improved our ability to support mammalian embryos throughout the preimplantation period entirely in vitro [1]. These advancements have enabled significant breakthroughs in both developmental biology research and assisted reproductive technologies (ARTs). With increasing sophistication in culture media formulations, current efforts are increasingly focused on mimicking the natural uterine environment to support optimal embryonic development and long-term health [2]. Among media components, nutritional supplements, particularly proteins, have emerged as key modulators of embryogenesis and metabolic programming [3]. A fundamental insight from developmental biology research is that environmental conditions during early embryogenesis can have lasting consequences extending far beyond the immediate developmental period [4]. This principle forms the basis of the developmental origins of health and disease (DOHaD) hypothesis, which posits that nutritional factors during sensitive developmental windows can permanently alter physiological systems [5]. The preimplantation period represents a critical window of developmental plasticity, during which the embryo is particularly vulnerable to nutritional perturbations that may disrupt epigenetic programming and cellular differentiation [6]. These early alterations can manifest as increased susceptibility to metabolic disorders and other health complications in later life [7]. Within this framework, the protein components of embryo culture media have become a major area of research [8]. In vivo, developing embryos are exposed to a complex mixture of proteins in oviductal and uterine fluids that serve multiple essential functions [9]. Among these, serum albumin has been identified as particularly crucial for supporting optimal embryonic development [10]. The most abundant protein in embryo culture media, BSA, a commonly used form of albumin, performs diverse physiological roles, including maintenance of osmotic pressure, pH regulation, lipid solubilization, and molecular transport [11,12]. These multifaceted functions make BSA an indispensable component of culture systems designed to support embryonic development outside the reproductive tract [13]. Efforts to develop completely defined, protein-free culture systems have consistently demonstrated the unique importance of BSA [14]. While synthetic macromolecules can partially substitute for some physical properties of BSA, they fail to replicate its full spectrum of biological activities [15]. This limitation highlights the complexity of BSA’s roles and underscores the need for better understanding of its developmental stage-specific contributions [16]. Despite decades of research demonstrating the importance of BSA supplementation for preimplantation embryo development [10,11,12,13,17], fundamental questions remain regarding the molecular and cellular mechanisms by which its absence affects embryo quality during critical periods of cell proliferation and differentiation. The current study systematically examined the effects of BSA deprivation on mouse embryo development. Our comprehensive approach evaluated developmental competence under different BSA deprivation regimens, quantified impacts on cell lineage specification, and assessed long-term metabolic outcomes in offspring. These findings provide new insights into preimplantation developmental biology while having practical implications for optimizing assisted reproductive technologies. By elucidating the consequences of BSA deprivation at specific developmental windows, our results contribute to efforts to minimize potential long-term health risks for offspring conceived through these procedures.

## 2. Results

### 2.1. Short-Term Effects of BSA Absence: Blastocyst Development Under BSA-Free Culture at Varying Time Points and Durations

As illustrated in Figure 1, the effect of BSA deprivation during preimplantation development was assessed by comparing blastocyst formation rates under different culture conditions.

As shown in Table 1, embryos cultured continuously in CZB medium (control group) showed the highest blastocyst development rate (97 ± 2.4%), establishing a baseline for comparison. Complete BSA deprivation throughout the 96 h culture period CZB-BSA(96h) group resulted in a statistically significant reduction in blastocyst formation (91 ± 7.4%; *p* < 0.05), as determined by the Wilcoxon each-pair test. In contrast, early-stage BSA deprivation groups (CZB-BSA(E24h) and CZB-BSA(E48h)) maintained developmental rates comparable to controls (99 ± 3.7% and 99 ± 4.9%, respectively). Groups with BSA deprivation after 24, 48, or 72 h (CZB-BSA(24h–96h), CZB-BSA(48h–96h), and CZB-BSA(72h–96h)) demonstrated consistently normal blastocyst rates (95 ± 8.0%, 96 ± 6.8%, and 95 ± 7.0%, respectively), showing no significant differences from the control group. These findings indicate that while prolonged BSA deprivation significantly impairs blastocyst formation, short-term absence does not substantially affect developmental progression to the blastocyst stage.

To evaluate blastocyst quality, cell lineage differentiation into trophectoderm (TE) and inner cell mass (ICM) was assessed via immunostaining using CDX2 and NANOG, respectively (Figure 2A,B). Embryos cultured without BSA throughout the entire culture period (CZB-BSA(96h)) exhibited significantly lower total cell numbers, TE, and ICM cell counts compared to the control group (Figure 2C–E). Specifically, the number of CDX2-positive TE cells was significantly reduced (Figure 2D) in embryos subjected to early BSA deprivation, as seen in CZB-BSA(E24h) and CZB-BSA(E48h) groups (38.4 ± 11.5 and 40.9 ± 13.4, respectively), as well as in the continuously deprived CZB-BSA(96h) group (41.4 ± 10.2), all significantly lower than controls (*p* < 0.05). Similarly, NANOG-positive ICM cell numbers were markedly diminished in the CZB-BSA(96h), CZB-BSA(E24h), and CZB-BSA(E48h) groups (10.5 ± 2.7, 9.6 ± 2.2, and 9.6 ± 1.9, respectively) (Figure 2E). Additionally, the CZB-BSA(72h–96h) group, which experienced BSA deprivation from the morula stage onward, also showed a significant decrease in ICM cell numbers (10.2 ± 3.1; *p* < 0.05 vs. control). In contrast, embryos in the CZB-BSA(48h–96h) group—exposed to BSA deprivation only after the four-cell stage—retained both TE and ICM cell numbers comparable to the CZB control. Furthermore, no statistically significant differences were observed in the ICM-to-TE ratio among groups (Appendix A).

### 2.2. Long-Term Effects on CZB- and CZB-BSA-Derived Offspring on Body Weight

To explore the effects of BSA absence in the culture medium on postnatal outcomes, embryos were cultured to the morula stage and transferred to pseudo-pregnant mice. As shown in Table 2, there was no discernible difference in the birth rates of the CZB and CZB-BSA groups, which were 40 ± 2.3 and 44 ± 2.9, respectively. At 19.5 days post-coitum (dpc), neonatal body weights showed no significant differences between experimental groups, with both CZB (2.4 ± 0.3 g) and CZB-BSA (2.3 ± 0.3 g) groups exhibiting comparable birth weights (*p* > 0.05).

This study investigated whether the absence of BSA in preimplantation culture media influences long-term metabolic development in mice. Weekly tracking of body weight (Figure 3) from birth to 16 weeks showed minimal effects on male mice. In male offspring, body weight at 8 weeks was comparable between the CZB-BSA group and the CZB control group (46.8 ± 3.8 g vs. 46.3 ± 5.1 g), with no statistically significant difference. By 16 weeks, CZB-BSA males exhibited a modest reduction in average body weight (53.8 ± 6.4 g) compared to CZB controls (57.1 ± 5.2 g), although this difference was not statistically significant (Student’s *t*-test, *p* > 0.05) (Figure 3B). A similar trend was observed in female offspring. At 8 weeks, body weight was comparable between CZB-BSA and CZB females (39.3 ± 4.0 g vs. 39.2 ± 4.2 g), but by 16 weeks, CZB-BSA females showed a slight reduction in body weight (47.8 ± 8.1 g) relative to CZB controls (50.4 ± 6.9 g), again without reaching statistical significance (Figure 3C).

### 2.3. Long-Term Effects of the Absence of BSA on Glucose Metabolism in Adult Mice

The impact of BSA deprivation during embryonic development on glucose metabolism was evaluated through longitudinal oral glucose tolerance tests (OGTTs) in offspring at 8 and 16 weeks of age. Statistical analysis was conducted using the Wilcoxon rank-sum test to evaluate glucose metabolism parameters between experimental and control groups. As shown in Figure 4, our findings reveal distinct temporal and sex-specific patterns of metabolic disruption associated with early BSA deprivation. Male offspring exposed to BSA-free conditions exhibited progressive glucose intolerance, while fasting glucose levels at 8 weeks were not significantly different than the control (*p* > 0.05; Figure 4A). However, by 16 weeks, BSA-deprived male offspring developed significant fasting hyperglycemia compared to control (*p* < 0.05; Figure 4C). OGTT profiles showed non-significant elevation in peak glucose levels at 15 min post-challenge at both ages (*p* > 0.05) followed by sustained hyperglycemia throughout the testing period, suggesting impaired glucose clearance capacity. Female offspring demonstrated preserved fasting glucose regulation at both 8 and 16 weeks (*p* > 0.05; Figure 4B,D). However, OGTT temporal analysis revealed a trend toward delayed glucose clearance between 60 and 120 min in 16-week-old females, although these differences did not reach statistical significance (*p* > 0.05). The iAUC analysis revealed significant sex-specific metabolic responses to embryonic BSA deprivation. Male offspring at 16 weeks exhibited substantially impaired glucose tolerance, demonstrating a 28% elevation in iAUC relative to controls (*p* < 0.05; Figure 4G). In contrast, female counterparts showed no statistically significant alteration in iAUC values (*p* > 0.05; Figure 4H), despite displaying similar trends.

## 3. Discussion

This study investigated the developmental consequences of BSA deprivation during early embryogenesis, focusing on its impact on blastocyst formation, cell lineage allocation, and long-term metabolic health. Our findings demonstrate that sustained BSA deprivation from the 2PN stage onward significantly impairs preimplantation development, likely due to progressive metabolic stress, as evidenced by reduced blastocyst formation rates across all protein-deprived groups. To the best of our knowledge, this is the first study to report the long-term metabolic effects of culturing IVF embryos in BSA-free conditions, specifically evaluating blood glucose regulation in the resulting offspring.

In embryo culture media, BSA serves multiple critical functions: as a protein supplement, a source of both amino acids and unidentified non-amino acid molecules that promote blastocyst development, and an additional chelating agent that enhances the base medium’s capabilities [18,19]. The significant decline in blastocyst formation rates in the CZB-BSA(96h) group underscores the detrimental effects of continuous BSA deprivation on embryonic competence. These results align with prior studies demonstrating that macromolecule-free culture conditions substantially hinder blastocyst development [15]. Notably, short-term BSA deprivation did not significantly impair development, indicating that preimplantation embryos can transiently adapt to temporary macromolecule scarcity. However, even short-term deprivation led to reduced TE and ICM cell counts in resulting blastocysts, consistent with previous findings on protein deficiency and embryo development [17,20]. Embryos deprived of BSA from early stages exhibited diminished cellular allocation to both TE and ICM lineages, supporting the paradigm that nutrient availability regulates metabolic pathways critical for lineage specification [21]. While late-stage BSA deprivation (CZB-BSA(72h–96h)) did not significantly affect TE or total cell numbers, it selectively reduced ICM formation, suggesting that post-compaction embryos have heightened protein requirements for fetal lineage development [22,23,24]. The ICM contributes to fetal tissues, and its selective reduction underlines a possible vulnerability in the developmental programming of the fetus, especially under nutrient-restricted conditions. Hence, BSA deprivation during this critical window may compromise developmental viability by impairing the establishment of fetal progenitor pools.

Although it is generally assumed that a minimum number of embryonic cells is necessary for pregnancy establishment [25], the optimal cell number and distribution between ICM and TE lineages remains unclear. While higher ICM cell counts correlate with improved pregnancy rates in some studies [26], others report superior outcomes in BSA-free cultures despite unaltered cell numbers [27]. This paradox extends to in vivo environments: sheep oviduct-derived embryos develop with fewer cells, yet maintain developmental competence compared to in vitro counterparts [28]. Similarly, protein-free culture systems show reduced miscarriage rates despite morphological differences [22], challenging the predictive value of conventional cell-based metrics. Our findings demonstrate that embryos cultured without BSA exhibit reduced TE and ICM cell numbers, yet achieved birth rates comparable to controls. This suggests a degree of developmental plasticity whereby embryos can compensate for early suboptimal conditions and still achieve successful implantation and development. Notably, embryos with lower ICM grades can still result in successful pregnancies when transferred into supportive uterine environments [29,30,31]. Lim et al. (2007) further demonstrated that bovine embryos cultured in defined media (supplemented with growth factors instead of BSA) showed comparable blastocyst cell counts, but higher pregnancy rates than BSA-containing media [27]. Collectively, these findings support the view that reduced cellularity does not necessarily compromise full-term developmental potential and that multiple factors beyond early morphology contribute to live birth outcomes [29].

Embryos with reduced TE and ICM cell numbers may also undergo compensatory catch-up growth post-transfer, resulting in normalized or even elevated fetal weights despite early cellular deficits. This developmental plasticity has been documented in both mouse and human ART models, where blastocysts with suboptimal morphology can still achieve robust fetal growth trajectories [32,33]. Consistently with this, our study found that CZB-BSA-derived offspring exhibited comparable birth weights to CZB controls, despite originating from blastocysts with significantly fewer cells. It is important to note that both CZB and CZB-BSA embryos were cultured in vitro, an inherently stressful environment that can induce adaptive changes in early programming. Previous studies have reported that IVF-derived mouse offspring frequently show higher birth weights than naturally conceived controls—a phenomenon attributed to metabolic programming [34]. Parallel findings in human ART cohorts have revealed associations between increased birth weights and elevated risks of adult-onset metabolic disorders, including type 2 diabetes and cardiovascular disease [35,36].

Our long-term data indicate that while no significant glucose intolerance was observed at 8 weeks, male CZB-BSA offspring exhibited impaired glucose tolerance by 16 weeks of age, as demonstrated by a significantly elevated iAUC. These findings point to persistent metabolic disruption initiated during early development. This suggests that early BSA deprivation not only influences immediate developmental competence but can also program systemic physiological outcomes in later life. These findings align with prior reports linking nutrient perturbation during early development to later-life dysregulation of glucose homeostasis [7,16]. The reduced TE cell numbers in the CZB-BSA group may have potentially impacted placental progenitor allocation, although we did not assess placental structure or function in this study. While prior studies link early disruptions in trophoblast development to altered nutrient delivery and long-term metabolic dysfunction [37,38], our data alone cannot confirm placental insufficiency. The observed reduction in TE cells, the origin of placental lineages, suggests a possible impact on placental development, but further studies directly examining placental structure and function are needed to validate this hypothesis. Thus, our results suggest a possible placental-mediated contribution to later glucose dysregulation. We acknowledge this as a limitation of the current study.

The glucose intolerance observed in males may stem from epigenetic modifications induced by protein-free culture conditions, affecting pathways critical for metabolic programming [39]. Such outcomes are consistent with observations in ART, where suboptimal culture conditions correlate with increased metabolic dysfunction in adulthood [40]. Interestingly, female offspring did not exhibit significant glucose intolerance, suggesting sex-specific resilience to early nutritional stress, possibly due to hormonal or metabolic regulatory differences [41]. The observed sex-specific postnatal differences in our study may reflect inherent differences in male and female embryo metabolism and developmental adaptability, as reported in mammalian embryos under varying culture conditions [42]. While our study focused on preimplantation protein deprivation during in vitro culture, similar long-term sex-specific outcomes have been observed following maternal periconceptional low-protein diets in vivo, which program cardiovascular and behavioral phenotypes in offspring [43,44]. However, the reduced body weight in female CZB-BSA offspring at 16 weeks implies that preimplantation factors may still influence growth trajectories in a sex-dependent manner [45].

While BSA is widely used to support embryo development, some preparations may contain impurities that exert cytotoxic effects. Evidence suggests that impurities such as endotoxins or fatty acid contaminants in crude BSA may compromise developmental potential. Moreover, commercial BSA is a complex mixture containing various bioactive components and potential contaminants, which could influence embryonic development and offspring outcomes beyond the absence of albumin alone. Recent findings suggest that using deionized BSA can reduce this risk and improve developmental outcomes, highlighting the potential for refining standard culture media to enhance embryo viability [46]. Future studies should identify which specific components of BSA are critical for supporting embryo development and preventing adverse long-term phenotypes. Additionally, comparing CZB medium with other clinically used culture systems, such as G1/G2 sequential media supplemented with human serum albumin (HSA), may clarify whether media formulation influences developmental competence and offspring health [47].

Collectively, these findings support the DOHaD hypothesis, wherein early environmental perturbations predispose individuals to metabolic disorders later in life [3,48]. The glucose intolerance observed in CZB-BSA males illustrates a biological trade-off: embryonic adaptation to nutrient stress may permit survival and birth, but it comes at a long-term physiological cost, particularly in metabolic health.

This study provides novel insights into the developmental and long-term metabolic consequences of BSA deprivation; however, certain limitations should be acknowledged. Our metabolic assessments focused on glucose tolerance. Future evaluations of insulin sensitivity and lipid profiles would provide a more comprehensive understanding. Embryos were transferred at the morula stage to maximize transferable numbers, as prolonged culture without BSA reduced blastocyst formation rates. However, transferring at the blastocyst stage in future studies may reveal more pronounced long-term effects. Although the number of embryos transferred per recipient varied, our analyses accounted for individual pup data. Nonetheless, standardizing embryo numbers in future studies is recommended. Additionally, although analyses of embryo quality markers (e.g., apoptosis, mitochondrial function) and sex-specific differences at the preimplantation stage would provide valuable mechanistic insights, they were not included within the scope of this study. Such assessments could have clarified the observed reductions in cell numbers and developmental competence. Given the sex-specific differences observed in postnatal metabolic outcomes, future studies incorporating embryo sexing will be valuable to determine whether early developmental responses differ between male and female embryos. Furthermore, assessments under high-fat dietary conditions, blood pressure monitoring, and comprehensive molecular analyses, including gene expression profiling, apoptotic index, mitochondrial function, and analysis of epigenetic changes, will be essential to elucidate the mechanistic impact of BSA deprivation during preimplantation development. Such investigations may clarify how the absence of BSA alters early embryonic programming and predisposes offspring to metabolic dysregulation.

## 4. Materials and Methods

### 4.1. Animals

Female and male ICR-strain mice aged between 8 and 12 weeks were procured from Shizuoka Laboratory Animal Center (SLC) Inc. (Hamamatsu, Japan). These mice were maintained in a specific pathogen-free (SPF) facility under controlled environmental conditions: temperature at 25 °C, relative humidity at 50%, and a 14–10 h light–dark cycle. They were provided ad libitum access to a standard pelleted diet and distilled water. All animal experiments were conducted in compliance with ethical guidelines and approved by the Animal Experimentation Committee at the University of Yamanashi, Japan, under protocol number A4–10.

### 4.2. Reagents

Unless specified otherwise, all reagents were procured from Sigma-Aldrich Chemical Co. (St. Louis, MO, USA), Fujifilm Wako Pure Chemical Co. (Osaka, Japan), or Nacalai Tesque Inc. (Kyoto, Japan).

### 4.3. In Vitro Fertilization (IVF)

IVF was conducted using the method described in a previous study [49]. To induce superovulation, female ICR mice (≥8 weeks old) were intraperitoneally injected with 7.5 IU of pregnant mare serum gonadotropin (PMSG; ASKA Pharmaceutical Co., Tokyo, Japan), followed 48 h later by 7.5 IU of human chorionic gonadotropin (hCG; ASKA Pharmaceutical Co.). At 14–17 h after hCG injection, cumulus–oocyte complexes (COCs) were collected from the oviducts into fertilization drops containing HTF medium [50]. Male ICR mice (≥10 weeks old) were euthanized by cervical dislocation, and cauda epididymides were dissected and punctured to release spermatozoa into HTF medium. Sperm were capacitated at 37 °C in an atmosphere of 5% CO_2_ for 30 min. Sperm concentration was assessed manually using a hemocytometer and adjusted to approximately 1.0–1.5 × 10^6^ sperm/mL. The diluted sperm suspension was added to the HTF droplet containing COCs, and insemination was carried out for 5–6 h. During this co-incubation period, denudation of the oocytes occurred naturally. After insemination, oocytes were washed to remove excess sperm and debris, and only zygotes showing the presence of two pronuclei and the second polar body (observed under an inverted microscope; IX71, Olympus, Tokyo, Japan) were selected for further culture. These were transferred to Chatot–Ziomek–Bavister (CZB) medium [51] with or without BSA, referred to as CZB and CZB-BSA groups, respectively. For CZB medium preparation, BSA fraction V, crystalline grade (12657-5GM; EMD Millipore Corp., Burlington, MA, USA; a subsidiary of Merck KGaA, Darmstadt, Germany) was used. All embryo cultures were maintained at 37 °C in a humidified atmosphere of 5% CO_2_ in air (approximately 20% O_2_).

### 4.4. Experimental Groups

To investigate the effects of BSA deprivation during preimplantation development, seven experimental groups were used to investigate the effects of the absence of BSA during the preimplantation stage in CZB culture media. The CZB group served as the control, wherein in vitro-fertilized embryos were cultured in CZB culture media from the 2PN stage to the blastocyst stage. The remaining six groups underwent culture without BSA in a time-dependent manner. The CZB-BSA(96h) group was created by culturing in vitro-fertilized embryos in CZB-BSA culture media from the 2PN stage to the blastocyst stage. For the CZB-BSA(E24h) group, embryos were cultured without BSA from the 2PN stage, and after 24 h, they were transferred to CZB culture media and observed until the blastocyst stage. Creation of the CZB-BSA(E48h) group involved culturing embryos in CZB-BSA culture media from the 2PN stage to the 4-cell stage, then transferring them to CZB culture media for 48 h before observation until the blastocyst stage. The CZB-BSA(24h–96h), CZB-BSA(48h–96h), and CZB-BSA(72h–96h) groups were cultured in CZB culture media for 24 h, 48 h, and 72 h, respectively, after the 2PN stage. Subsequently, they were transferred to CZB-BSA culture media and observed until the blastocyst stage. All experimental groups are shown in Figure 1.

### 4.5. Immunostaining

Immunostaining procedures were conducted following the methods outlined in Fulka and Langerova [52], with minor adjustments. Briefly, embryos were washed twice in PBS supplemented with 1% polyvinyl alcohol (PBS–PVA) and were then fixed in 4% paraformaldehyde (PFA) in PBS for 20 min at room temperature. Following fixation, embryos were washed in PBS–PVA and incubated overnight at 4 °C in blocking buffer consisting of 0.1% Triton X-100 and 1% BSA in PBS. Visualization of nucleoli was achieved using the mouse monoclonal anti-NOPP140 antibody (dilution 1:500, sc-374,033; Santa Cruz Biotechnology, Inc., Santa Cruz, CA, USA). To identify the inner cell mass (ICM) and trophectoderm (TE) cells of blastocysts, primary antibodies included an anti-NANOG rabbit polyclonal antibody (dilution 1:500, ab80892; Abcam, Cambridge, UK) for ICM and an anti-CDX2 mouse monoclonal antibody (dilution 1:500, CM226; BioCare Medical, Pacheco, CA, USA) for TE. After further washes, secondary antibodies (Alexa Fluor 568-conjugated goat anti-mouse IgG, 488-conjugated goat anti-rabbit IgG) were added, and embryos were immersed for 2 h at 4 °C. Finally, embryos were mounted on glass slides in Vecta-shield (Vector Laboratories Inc., Burlingame, CA, USA) supplemented with 1 µg/mL 4′,6-diamidino-2-phenylindole (DAPI). The images were obtained with a Keyence-800 microscope. Quantitative analysis from these images for the number of cells was performed using Image J software (version 1.53t, National Institutes of Health, Bethesda, MD, USA).

### 4.6. Embryo Transfer

For the long-term experiments, two experimental groups were established: CZB and CZB-BSA. In the CZB-BSA group, in vitro-fertilized embryos were cultured in BSA-free CZB medium from the two-pronuclear (2PN) stage through to the morula stage, whereas embryos in the CZB group were cultured in standard CZB medium during the same developmental window. Embryo transfer was conducted as previously described [53], with minor modifications. In summary, two sets of embryos were employed. At 0.5 dpc, following mating with vasectomized ICR male mice, CZB-BSA embryos, cultured in the absence of BSA from 2PN stage to the morula stage, and CZB embryos, cultured in CZB culture media from 2PN stage to the morula stage, were transferred into oviducts of pseudopregnant ICR female mice. Under anesthesia induced by intraperitoneal injection of ketamine (10 mg/100 g body weight; Ketasol, Graeub Veterinary Products) and xylazine (0.4 mg/100 g body weight; Rompun, Graeub Veterinary Products), 6–10 embryos were transferred into each oviduct of recipient females. The number of embryos transferred varied slightly among recipients due to differences in embryo availability from each experimental group. This approach aimed to maximize use of available embryos while ensuring viable pregnancies, as practiced in similar embryo transfer studies. A natural delivery occurred at 19.5 dpc, and body weight measurements were made every week after delivery. After weaning at 3 weeks, male and female pups were separated and body weight monitoring continued until 16 weeks of age.

### 4.7. Oral Glucose Tolerance Test (OGTT)

To evaluate long-term metabolic effects of preimplantation culture conditions, an oral glucose tolerance test (OGTT) was performed on CZB and CZB-BSA derived offspring at both 8 and 16 weeks of age. Mice were fasted for 6 h prior to the test, with access to water maintained throughout the fasting period. Following the fasting period, baseline blood glucose levels were measured using a glucometer from tail vein blood. Mice were then administered a glucose solution (1 g/kg body weight) orally via gavage. Subsequent blood glucose measurements were taken at 15, 30, 60, and 120 min post-glucose administration. Glucose tolerance curves were generated for each group, and the incremental area under the curve (iAUC) was calculated to assess glucose handling efficiency over time.

### 4.8. Statistical Analysis

Statistical analyses were performed using JMP Pro version 16.0 (SAS Institute Inc., Cary, NC, USA). The Wilcoxon rank-sum test and Student’s *t*-test were applied as appropriate to compare blastocyst cell counts and offspring body weights between experimental groups. The rates of blastocyst formation and birth outcomes were evaluated with Fisher’s exact test. A *p*-value of less than 0.05 was considered statistically significant. Graphical representations were generated using both JMP Pro and RStudio (Version 2025.05.0 + 496; RStudio, PBC, Boston, MA, USA).

## 5. Conclusions

This study demonstrates that deprivation of BSA during preimplantation embryo culture impairs blastocyst formation and cell lineage allocation, but does not prevent implantation or birth, indicating a degree of developmental plasticity. However, embryos cultured without BSA exhibited long-term sex-specific metabolic disturbances, particularly impaired glucose tolerance in adult males. These findings suggest that early protein availability influences not only immediate developmental competence but also postnatal metabolic programming. Collectively, our data support the DOHaD hypothesis and highlight the need for careful optimization of culture media components to promote both embryonic development and long-term offspring health.

## Figures and Tables

**Figure 1 ijms-26-06989-f001:**
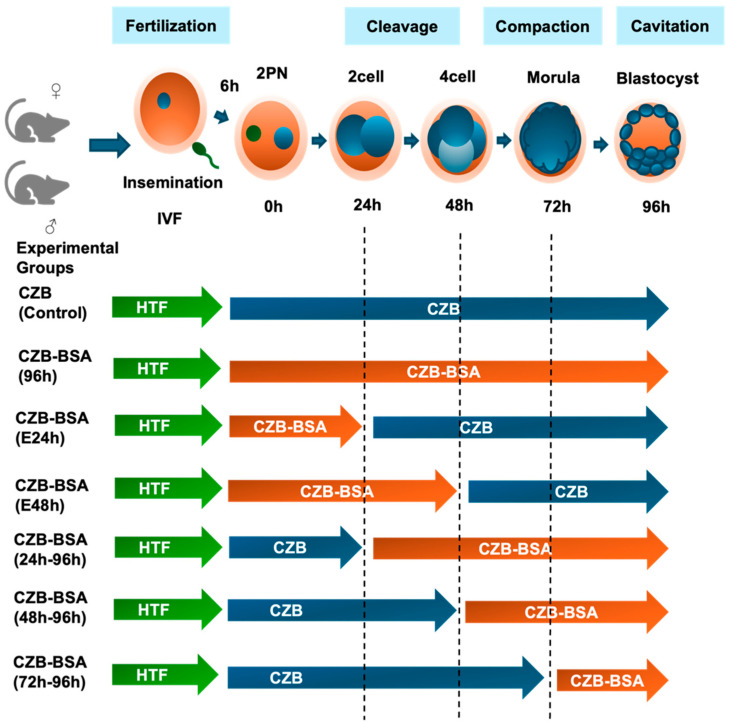
Experimental design illustrating embryo culture conditions following IVF. Zygotes (2PN) were cultured in CZB medium with or without BSA to assess the effect of protein deprivation during specific preimplantation stages. All groups were fertilized in HTF medium and then cultured under different time windows in CZB (with BSA, blue arrows) or CZB-BSA (without BSA, orange arrows). The control group remained in CZB for 96 h, while experimental groups were exposed to CZB-BSA for the full 96 h or specific intervals (0–24 h, 0–48 h, 24–96 h, 48–96 h, or 72–96 h). Developmental stages are marked to indicate media transitions.

**Figure 2 ijms-26-06989-f002:**
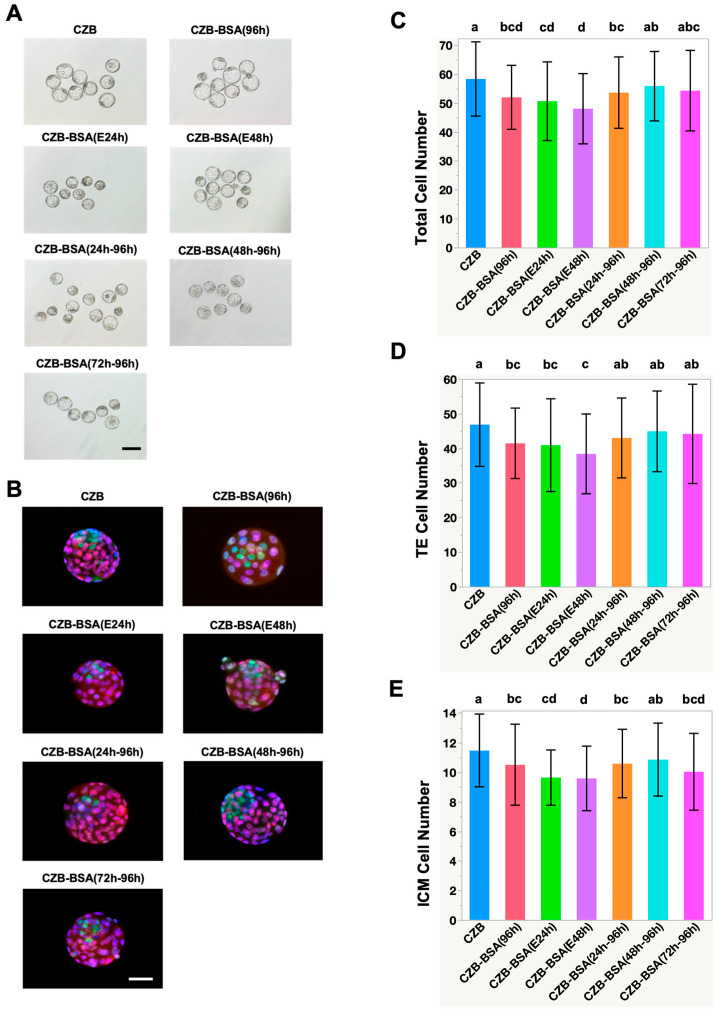
Effect of BSA deprivation at different developmental time windows on blastocyst morphology and cell lineage allocation. (**A**) Representative bright-field images of expanded blastocysts from each experimental group at 96 h post-insemination. Embryos were cultured either continuously in CZB medium (CZB) or in BSA-deprived CZB-BSA medium for various durations: CZB-BSA(96h) for the full duration, E24h and E48h for early-stage deprivation, CZB-BSA(24h–96h) and CZB-BSA(48h–96h) for mid- to late-stage deprivation, and CZB-BSA(72h–96h) for late-stage deprivation only. Scale bar = 50 µm. (**B**) Representative immunofluorescence images of blastocysts stained with DAPI (blue, nuclei), CDX2 (red, trophectoderm marker), and NANOG (green, inner cell mass marker). (**C**–**E**) Quantification of total cell numbers (top panel), TE cell numbers (middle panel), and ICM cell numbers (bottom panel) in blastocysts from each group (*n* ≥ 40). Different superscript letters indicate significant differences between groups based on pairwise Wilcoxon rank-sum tests (*p* < 0.05). Groups sharing the same letter are not significantly different.

**Figure 3 ijms-26-06989-f003:**
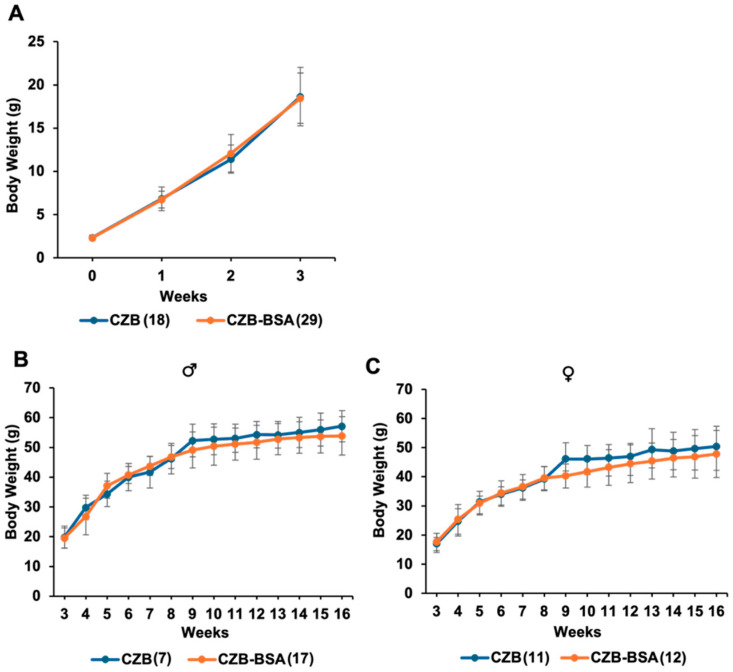
Postnatal body weight of CZB- and CZB-BSA-derived offspring. (**A**) Average body weight of pups from birth to 3 weeks of age (*n* = 20–30 pups/group). (**B**) Weekly body weight progression of male offspring from 3 to 16 weeks of age. (**C**) Weekly body weight progression of female offspring from 3 to 16 weeks of age. Numbers in parentheses indicate the number of animals in each group used for body weight measurements. Error bars represent the SEM.

**Figure 4 ijms-26-06989-f004:**
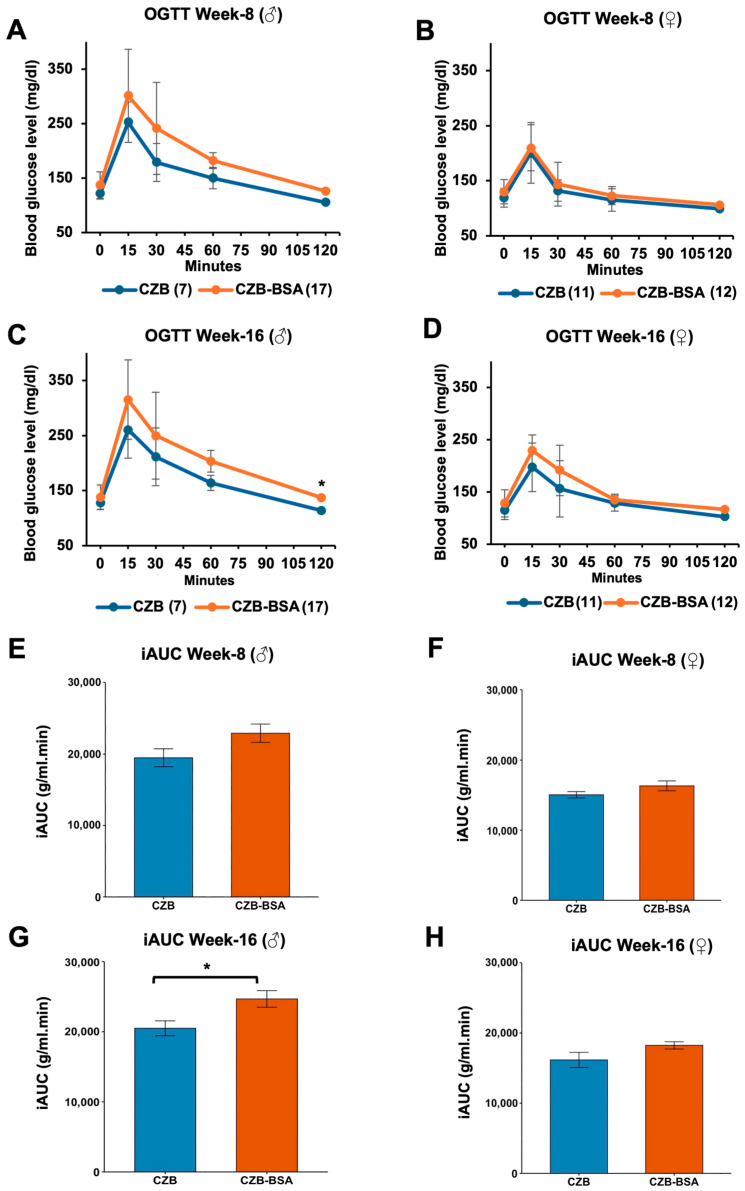
OGTT and glucose clearance in CZB- and CZB-BSA-derived offspring. (**A**,**B**) Blood glucose levels measured at 0, 15, 30, 60, and 120 min post-glucose administration in male (**A**) and female (**B**) offspring at 8 weeks of age. (**C**,**D**) Glucose clearance profiles in male (**C**) and female (**D**) offspring at 16 weeks of age. (**E**–**H**) Mean incremental area under the curve (iAUC) values for males at 8 weeks (**E**) and 16 weeks (**G**) and for females at 8 weeks (**F**) and 16 weeks (**H**). Numbers in parentheses indicate the number of animals in each group. The Wilcoxon rank-sum test was used to assess statistical significance. Asterisks (*) indicate a significant difference (*p* < 0.05). Data are shown as means ± SEM. Gray error bars represent the SEM.

**Table 1 ijms-26-06989-t001:** Blastocyst development rates under BSA-free culture conditions across varying preimplantation time windows.

Experimental Group	No. of Embryos Developed (Means of Percentages ± SEM)				
	2PN	2-Cell	4-Cell	Morula	Blastocyst
CZB (Control)	127	127 (100 ± 0)	127 (100 ± 0)	127 (100 ± 0)	125 (97 ± 2.4)
CZB-BSA(96h)	138	138 (100 ± 0)	135 (98 ± 4.9)	133 (97 ± 5.7)	124 (91 ± 7.4) *
CZB-BSA(E24h)	53	53 (100 ± 0)	53 (100 ± 0)	53 (100 ± 0)	51 (99 ± 3.7)
CZB-BSA(E48h)	90	90 (100 ± 0)	90 (100 ± 0)	89 (99 ± 4.4)	89 (99 ± 4.9)
CZB-BSA(24h–96h)	94	94 (100 ± 0)	93 (98 ± 3.6)	92 (98 ± 7.5)	89 (95 ± 8.0)
CZB-BSA(48h–96h)	106	106 (100 ± 0)	106 (100 ± 0)	105 (99 ± 2.7)	101 (96 ± 6.8)
CZB-BSA(72h–96h)	86	86 (100 ± 0)	86 (100 ± 0)	86 (100 ± 0)	82 (95 ± 7.0)

Note: Blastocyst development rates were assessed by dividing the period from the preimplantation period into cultures of varying time durations, all conducted in the absence of BSA. Data are expressed as means ± SEM. Asterisk (*) indicates statistically significant differences between conditions. A *p*-value of <0.05 indicates statistically significant differences between conditions (*p* < 0.05), highlighting the impact of BSA exposure timing on embryo viability.

**Table 2 ijms-26-06989-t002:** Birth rate following preimplantation culture in BSA-free conditions.

Experimental Group	No. of Transferred Embryos	No. of Recipients	No. of Live Offspring	Means of Birth Rate % ± SEM	Average Body Weight at Birth (19.5 Days) g ± SEM
CZB	63	6	25	40 ± 2.3	2.4 ± 0.3
CZB-BSA	93	7	41	44 ± 2.9	2.3 ± 0.3

Note: No significant statistical difference.

## Data Availability

The data presented in this study are available on request from the authors.

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
