# Peer review of "The Absence of Bovine Serum Albumin (BSA) in Preimplantation Culture Media Impairs Embryonic Development and Induces Metabolic Alterations in Mouse Offspring"

_ijms, 2025, doi:10.3390/ijms26146989_

Round 1

Reviewer 1 Report

Comments and Suggestions for Authors

IJMS – manuscript review, June 2025 ijms-3713100

Title: The absence of bovine serum albumin (BSA) in preimplantation culture media impairs embryonic development and induces metabolic alterations in mouse offspring

Corresponding author: Satoshi Kishigami

The authors sought to investigate the effect of removing BSA from preimplantation embryo culture, observing that BSA removal affected cell lineage allocation and negatively affected metabolic health of resulting male offspring. Overall, the manuscript is very well written and easy to read. However, I have several concerns that need to be addressed:

  • What was the rational for culturing the embryos in atmospheric oxygen as opposed to 5% oxygen (physiological)? Supraphysiological oxygen regulates preimplantation embryo metabolism and amino acid turnover (PMID: 22553221).

  • Why wasn’t the same number of embryos transferred to each recipient mouse? How did you account for variations in litter size when performing statistical analysis on pup outcomes like weight?

  • What was the rational for transferring the embryos at morula stage instead of blastocyst stage? Surely transferring at the blastocyst stage (longer BSA deprivation) would have a more profound effect on pup outcomes.

  • Figure 2 C, D, E – it isn’t possible to see the average of each treatment underneath the individual data points in the graph. Please also include error bars.

  • Have you calculated the ICM to TE ratio? This is an important measure to distinguish changes in blastocyst cell number vs. cell lineage allocation.

  • I am not sure that you can claim a placental link without having examined the placenta. To say that there are fewer TE cells doesn’t conclusively mean the placenta is impaired. There was also significantly less ICM cells – perhaps BSA deprivation slows embryo growth, thus accounting for fewer cell numbers. If the ICM to TE ratio was different with BSA deprivation – fewer TE cells relative to ICM cells (reflective of a change in cell lineage allocation) – then I would be more inclined to accept a placental link.

  • It would be interesting to see how embryos cultured in CZB without and without BSA compare to embryos cultured in alternate culture medium systems, such as G1/G2 sequential media, which contains both non-essential and essential amino acids and is supplemented with HSA (rather than BSA).

Author Response

Comment 1: What was the rationale for culturing the embryos in atmospheric oxygen as opposed to 5% oxygen (physiological)? Supraphysiological oxygen regulates preimplantation embryo metabolism and amino acid turnover (PMID: 22553221).

Response: Thank you for this valuable comment. We cultured embryos in atmospheric oxygen (20% O₂) under 5% CO₂, following standard mouse embryo culture protocols in our laboratory and to ensure comparability with our previous studies. While we acknowledge that 5% O₂ better reflects physiological conditions (PMID: 22553221), all experimental groups were exposed to identical oxygen tension, minimizing differential effects in our results.

Comment 2: Why wasn’t the same number of embryos transferred to each recipient mouse? How did you account for variations in litter size when performing statistical analysis on pup outcomes like weight?

Response: We appreciate the reviewer’s observation. The number of embryos transferred varied due to availability from different culture groups and to maximize the use of recipient females while ensuring viable pregnancies. To account for litter size variations, offspring body weight and metabolic data were analyzed on an individual pup basis, independent of litter size, which is consistent with previous similar studies. However, we acknowledge that litter size can influence outcomes and will consider incorporating litter size as a covariate in future analyses to refine statistical accuracy.

Comment 3: What was the rational for transferring the embryos at morula stage instead of blastocyst stage? Surely transferring at the blastocyst stage (longer BSA deprivation) would have a more profound effect on pup outcomes.

Response: Thank you for this insightful question. Embryo transfer was performed at the morula stage because our primary aim was to evaluate the long-term effects of BSA deprivation specifically during pre-compaction development. Additionally, transferring at morula stage increased the number of transferable embryos across groups, as prolonged culture without BSA led to reduced blastocyst formation. We agree that blastocyst-stage transfers would provide valuable additional information and plan to examine this in follow-up studies.

Comment 4: Figure 2 C, D, E – it isn’t possible to see the average of each treatment underneath the individual data points in the graph. Please also include error bars.

Response: Thank you for pointing this out. We have revised Figure 2C–E to clearly display mean values and standard error bars for each group. The figure legend has also been updated accordingly.

Comment 5: Have you calculated the ICM to TE ratio? This is an important measure to distinguish changes in blastocyst cell number vs. cell lineage allocation.

Response: We thank the reviewer for this insightful suggestion. In response, we calculated the ICM to TE ratio for each experimental group to assess potential effects on cell lineage allocation. As presented in the newly added Supplementary Figure S1, our analysis showed no statistically significant differences in the ICM:TE ratio among the groups. This indicates that while BSA deprivation reduced total cell numbers (as shown in Figure 2C-E), it did not alter lineage allocation between the ICM and TE cells. These findings suggest that BSA deprivation primarily affects overall cell proliferation rather than the mechanisms governing cell fate specification.

Comment 6: I am not sure that you can claim a placental link without having examined the placenta. To say that there are fewer TE cells doesn’t conclusively mean the placenta is impaired. There was also significantly less ICM cells – perhaps BSA deprivation slows embryo growth, thus accounting for fewer cell numbers. If the ICM to TE ratio was different with BSA deprivation – fewer TE cells relative to ICM cells (reflective of a change in cell lineage allocation) – then I would be more inclined to accept a placental link.

Response: We acknowledge the reviewer’s concern. While our data showed reduced TE cell numbers, we agree that without direct placental examination, a definitive placental impairment conclusion cannot be drawn. We have revised the discussion to reflect this by suggesting a possible placental contribution rather than a confirmed link. Additionally, our ICM/TE ratio analysis indicated no significant change in lineage allocation, supporting that BSA deprivation primarily affects proliferation rather than lineage specification. We have modified the manuscript accordingly to avoid overstatement.

Comment 7: It would be interesting to see how embryos cultured in CZB without and without BSA compare to embryos cultured in alternate culture medium systems, such as G1/G2 sequential media, which contains both non-essential and essential amino acids and is supplemented with HSA (rather than BSA).

Response: Thank you for this valuable suggestion. We agree that comparing CZB with sequential media systems such as G1/G2 supplemented with HSA would strengthen the translational relevance of our findings. While such experiments were beyond the scope of the current study, we have added this point in the discussion as a limitation and future direction to assess whether culture media composition modulates the impact of protein deprivation on developmental and long-term outcomes.

Reviewer 2 Report

Comments and Suggestions for Authors

The manuscript by Jharna et. al., explores the developmental and metabolic consequences of bovine serum albumin (BSA) deprivation during the preimplantation stages in mice, offering important insights into the role of culture media components in early embryonic development and long-term metabolic programming. The study is relevant and timely, especially in the context of improving assisted reproductive technologies. However, several areas require further clarification and expansion to strengthen the conclusions:

Insufficient assessment of post-culture embryo quality: The authors primarily focus on blastocyst formation and cell numbers in the trophectoderm (TE) and inner cell mass (ICM). However, to comprehensively evaluate the impact of BSA deprivation, additional indicators of embryo quality—such as apoptotic index, mitochondrial activity, gene expression of pluripotency markers, or epigenetic modifications—should be incorporated. These metrics would help confirm the observed developmental impairments.

Lack of sex-specific mechanistic insights: While the study notes differential long-term metabolic outcomes in female offspring, it does not provide sufficient analysis of how male and female embryos are differently affected by BSA deprivation. The authors should explore sex-specific responses during preimplantation and postnatal stages, possibly by stratifying developmental and metabolic data by sex, or discussing known sex-related metabolic programming differences.

Limited scope of metabolic evaluations: The reported impairment in glucose clearance is an important observation, but it alone may not capture the full extent of metabolic dysfunction. The authors are encouraged to expand their metabolic assessment to include other parameters such as insulin sensitivity, lipid metabolism, and mitochondrial respiration, which could provide a more comprehensive understanding of the metabolic phenotype.

Uncertainty regarding the composition of BSA: Given that bovine serum albumin is a complex and not fully defined protein mixture, the manuscript should emphasize in the discussion that the observed effects may result from the absence of not just albumin itself but also other bioactive components or contaminants within commercial BSA preparations. Identifying or hypothesizing which specific components might be responsible for the observed phenotypes would significantly enhance the biological relevance of the findings.

Overall, this manuscript addresses an important topic but would benefit from deeper mechanistic insight, broader phenotypic analysis, and more nuanced interpretation of BSA’s composition and role. Addressing these points would considerably improve the scientific rigor and impact of the study.

Author Response

Comment 1: Insufficient assessment of post-culture embryo quality: The authors primarily focus on blastocyst formation and cell numbers in the TE and ICM. However, to comprehensively evaluate the impact of BSA deprivation, additional indicators of embryo quality- such as apoptotic index, mitochondrial activity, gene expression of pluripotency markers, or epigenetic modifications- should be incorporated. These metrics would help confirm the observed developmental impairments.

Response: We thank the reviewer for this constructive suggestion. Due to experimental limitations, we did not assess apoptotic index, mitochondrial activity, gene expression, or epigenetic modifications in this study. However, we agree that such analyses would provide deeper mechanistic insights and have noted this as an important limitation and future research direction in the revised discussion section.

Comment 2: Lack of sex-specific mechanistic insights: While the study notes differential long-term metabolic outcomes in female offspring, it does not provide sufficient analysis of how male and female embryos are differently affected by BSA deprivation. The authors should explore sex-specific responses during pre-implantation and postnatal stages, possibly by strartifying developmental and metabolic data by sex, or discussing known sex-related metabolic programming differences.

Response: We appreciate the reviewer highlighting this aspect. In our current dataset, preimplantation embryos were not sexed; however, long-term data were stratified by sex, revealing male-specific glucose intolerance. We have expanded the discussion to include known sex-related differences in metabolic programming, suggesting potential hormonal or epigenetic mechanisms underlying the observed male-specific phenotype.

Comment 3: Limited scope of metabolic evaluations: The reported impairment in glucose clearance is an imortant observation, but it alone may not capture the full extent of metabolic dysfunction. The authors are encouraged to expand their metabolic assessment to include other parameters such as insulin sensitivity, lipid metabolism, and mitochondrial respiration, which could provide a more comprehensive understanding of the metabolic phenotype.

Response: We agree with the reviewer that a comprehensive metabolic assessment would strengthen the study. Due to logistical constraints, our evaluations were limited to glucose tolerance. We have acknowledged this limitation in the discussion and highlighted plans to include insulin sensitivity, lipid profiling, and mitochondrial assessments in future studies to better define the metabolic consequences of early BSA deprivation.

Comment 4: Uncertainty regarding the composition of BSA: Given that bovine serum albumin is a complex and not fully defined protein mixture, the manuscript should emphasize in the discussion that the observed effects may result from the absence of not just albumin itself but also other bioactive components or contaminants within commercial BSA preparations. Identifying or hypothesizing which specific components might be responsible dor the observed phenotypes would significantly enhance the biological relevance of the findings.

Response: Thank you for this insightful comment. We have revised the discussion to emphasize that commercial BSA preparations may contain various bioactive components or contaminants (e.g. endotoxins, fatty acids) that could contribute to the observed effects. We also noted that identifying specific components responsible for these phenotypes is crucial for improving culture media formulations and plan to investigate this in future studies.

Reviewer 3 Report

Comments and Suggestions for Authors

Dear Satoshi Kishigami

Journal: International Journal of Molecular Sciences

Manuscript ID: ijms-3713100

Type: Article

Title:

The absence of bovine serum albumin (BSA) in preimplantation culture media impairs embryonic development and induces metabolic alterations in mouse offspring

Comments

In vitro fertilization (IVF)

  1. “sperm counts were assessed” Please clarify the procedures of counting and whether the count was performed manually or by CASA.
  2. Concentration on how authors obtain 1 × 106 sperm/mL, if not counted, please add approximately or about 1 × 106 sperm/mL.
  3. Before IVF, there is a missing technique that the authors did not mention. Please provide the procedures to obtain denuded oocytes COCs.
Comments on the Quality of English Language

The English could be improved to more clearly express the research.

Author Response

Comment 1: Sperm counts were assessed? Please clarify the procedures of counting and whether the count was performed manually or by CASA.

Response: Thank you for your question. We have clarified in the revised Materials and Methods section that sperm concentration was assessed manually using a hemocytometer under light microscopy following standard procedures, not by CASA.

Comment 2: Concentration on how authors obtain 1–1.5 × 10⁶ sperm/mL. If not counted, please add approximately of about 1–1.5 × 10⁶ sperm/mL.

Response: We appreciate this comment. In our revised Materials and Methods section, we now state that sperm concentration was manually calculated using a hemocytometer and adjusted to approximately 1.0–1.5 × 10⁶ sperm/mL before insemination.

Comment 3: Before IVF, there is a missing technique that the authors did not mention. Please provide the procedures to obtain denuded oocytes (COCs).

Response: Thank you for pointing this out. We have updated the Materials and Methods section to specify that COCs were co-incubated with sperm during the insemination period (5–6 hours), during which denudation occurred naturally. Following insemination, oocytes were washed to remove residual sperm and debris before transfer to CZB medium. No enzymatic denudation (e.g., hyaluronidase) was used.

Round 2

Reviewer 2 Report

Comments and Suggestions for Authors

The reviesd manuscript addresses a timely and important topic concerning the effects of BSA deprivation during early embryonic development and its long-term metabolic consequences. 
 However, while the current data provide valuable insights, I believe that the inclusion of additional experimental evidence—particularly regarding embryo quality (e.g., apoptotic index, mitochondrial function, or expression of pluripotency markers)—would substantially enhance the mechanistic strength of the study. These additions, even in a limited or preliminary form, would provide direct support for the developmental impairments described and better contextualize the long-term phenotypes observed. If feasible, retrospective sexing of embryos or stratification of preimplantation outcomes by sex would also be valuable, given the reported sex-specific postnatal effects. In addition, if these experiments are incorporated, I recommend that the authors revise the Introduction and Discussion sections.

Minor concern, 

-Clearly state in the Introduction the existing knowledge gaps regarding mechanistic consequences of BSA deprivation at the molecular and cellular levels, thus setting the stage for the inclusion of new embryo quality metrics.

-Expand the Discussion to integrate any newly generated data and more thoroughly reflect on how these findings support or refine current models of early nutritional programming.

-Provide a more nuanced view of the potential mechanisms underlying sex-specific outcomes, drawing on both the literature and any added experimental insights.

Taken together, these revisions would considerably improve the overall impact and clarity of the manuscript. Therefore, I recommend acceptance pending minor revision, with the suggestion to include targeted experimental additions and aligned textual updates in the Introduction and Discussion.

Author Response

We are grateful for the many valuable suggestions and comments in improving our manuscript. We truly appreciate the dedication the reviewer and those involved in improving our research and manuscript.

Reviewer 2

Comment: The reviesd manuscript addresses a timely and important topic concerning the effects of BSA deprivation during early embryonic development and its long-term metabolic consequences. 
 However, while the current data provide valuable insights, I believe that the inclusion of additional experimental evidence—particularly regarding embryo quality (e.g., apoptotic index, mitochondrial function, or expression of pluripotency markers)—would substantially enhance the mechanistic strength of the study. These additions, even in a limited or preliminary form, would provide direct support for the developmental impairments described and better contextualize the long-term phenotypes observed. If feasible, retrospective sexing of embryos or stratification of preimplantation outcomes by sex would also be valuable, given the reported sex-specific postnatal effects. In addition, if these experiments are incorporated, I recommend that the authors revise the Introduction and Discussion sections.

Response: We sincerely thank the reviewer for their insightful feedback and valuable suggestions to strengthen our manuscript. We fully agree that assessing embryo quality through additional metrics would further enhance the mechanistic interpretation of our findings.

            We would like to highlight that our study already includes quantitative analysis of key pluripotency markers (NANOG and CDX2), which serve as well-established indicators of embryonic cell lineage specification and developmental potential (Figure 2). As demonstrated in previous studies (Chambers et al., 2003; Silva et al., 2009), NANOG expression specifically marks pluripotent inner cell mass (ICM) cells during blastocyst formation, making it a robust marker for evaluating lineage specification and embryo quality. Our data on NANOG and CDX2 expression, combined with blastocyst rates and cell number quantification, provide direct evidence of how BSA deprivation impacts early embryonic development at the cellular level.

           While we acknowledge that additional analyses (e.g., apoptotic index, mitochondrial function, or embryo sexing) could offer deeper mechanistic insights, technical limitations and facility constraints prevented us from performing these specific experiments in the current study. However, to address the reviewer's concerns as thoroughly as possible, we have now revised the Introduction to clearly state the existing knowledge gaps at molecular and cellular levels and expanded the Discussion to acknowledge these limitations and propose these analyses as important future directions to elucidate the mechanisms of BSA deprivation effects. We also added a more nuanced discussion of potential sex-specific mechanisms underlying the observed postnatal differences. We hope these revisions address the reviewer’s comments satisfactorily.

Minor concerns

Comment: Clearly state in the Introduction the existing knowledge gaps regarding mechanistic consequences of BSA deprivation at the molecular and cellular levels, thus setting the stage for the inclusion of new embryo quality metrics.

Response: We have revised the Introduction to more explicitly highlight the current gaps in understanding the molecular and cellular mechanisms by which BSA deprivation affects embryo development and long-term outcomes.

Comment: Expand the Discussion to integrate any newly generated data and more thoroughly reflect on how these findings support or refine current models of early nutritional programming. Provide a more nuanced view of the potential mechanisms underlying sex-specific outcomes, drawing on both the literature and any added experimental insights.

Response: We have expanded the Discussion to emphasize our findings in the context of current models of early nutritional programming and discussed possible mechanisms underlying the sex-specific postnatal effects, referencing relevant literature. We have also highlighted the need for future studies incorporating apoptosis, mitochondrial function, and embryo sexing analyses to further clarify these mechanisms.